META-RESEARCH ARTICLE

# A qualitative study of the barriers to using blinding in in vivo experiments and suggestions for improvement

**Natasha A. Karp**[1]*, **Esther J. Pearl**[2], **Emma J. Stringer**[3], **Chris Barkus**[2], **Jane Coates Ulrichsen**[4], **Nathalie Percie du Sert**[2]

**1** Data Sciences & Quantitative Biology, Discovery Sciences, R&D, AstraZeneca, Cambridge, United Kingdom, **2** NC3Rs, London, United Kingdom, **3** Biomedical Services Unit, University of Birmingham, Birmingham, United Kingdom, **4** Early Oncology, AstraZeneca, Cambridge, United Kingdom

* natasha.karp@astrazeneca.com

**Data Availability Statement:** In accordance with our ethical permissions, for each experiment, the data (study type, randomisation strategy, the

## Abstract

In animal experiments, blinding (also known as masking) is a methodological strategy to reduce the risk that scientists, animal care staff, or other staff involved in the research may consciously or subconsciously influence the outcome. Lack of masking has been shown to correlate with an overestimation of treatment efficacy and false positive findings. We conducted exploratory interviews across academic and a commercial setting to discuss the implementation of masking at four stages of the experiment: during allocation and intervention, during the conduct of the experiment, during the outcome assessment, and during the data analysis. The objective was to explore the awareness, engagement, perceptions, and the barriers to implementing masking in animal experiments. We conducted multiple interviews, to explore 30 different experiments, and found examples of excellent practice but also areas where masking was rarely implemented. Significant barriers arose from the operational and informatic systems implemented. These systems have prioritised the management of welfare without considering how to allow researchers to use masking in their experiments. For some experiments, there was a conflict between the management of welfare for an individual animal versus delivering a robust experiment where all animals are treated in the same manner. We identified other challenges related to the level of knowledge on the purpose of masking or the implementation and the work culture. The exploration of these issues provides insight into how we, as a community, can identify the most significant barriers in a given research environment. Here, we offer practical solutions to enable researchers to implement masking as standard. To move forward, we need both the individual scientists to embrace the use of masking and the facility managers and institutes to engage and provide a framework that supports the scientists.

## Introduction

Masking (also known as blinding) is a methodological process where the allocation to an experimental group (a group of test subjects that receives the same intervention in an

masking level achieved and the associated barriers) are provided in supplementary files.

**Funding:** The authors received no specific funding for this work.

**Competing interests:** I have read the journal's policy and the authors of this manuscript have the following competing interests: JCU and NAK have shareholdings in AstraZeneca. EJP, CB and NPdS are NC3Rs staff; role includes promoting the ARRIVE guidelines and the Experimental Design Assistant (EDA). EJS is a former NC3Rs staff member.

experiment) is concealed from the people running the experiment or analysing the data, to minimise subconscious bias and maximise the validity of the results. In this paper, we choose to refer to this process as "masking," which has not been used as frequently as "blinding" in the context of in vivo experiments, but there has been a move away from the term "blinding" as it associates being blind with lacking knowledge and perpetuates negative stereotypes about blind people [1]. Masking has been shown to stop the expectations of the researcher from unintentionally affecting the experiment or interpretation of the data, typically to support the preferred hypothesis. For example, in rat studies, particular expectations regarding the animals' performance in different experimental groups can lead the researchers to subconsciously handle the animals differently and create spurious results [2]. Researchers' expectations can also influence the observations themselves. In a study of pig behaviour where the same video was altered slightly and shown to researchers twice but with different labels, researchers were shown to score identical behaviours differently, based on priming through the video label [3]. Similar effects have been replicated in studies with a variety of species (e.g., in cattle, chickens, ants, and flat worms) [3–5]. Multiple systematic reviews of animal and preclinical studies have found that when outcome assessments are not masked, the intervention effect is overestimated. For example, a study of in vivo and cell-based experiments found that those that are neither masked nor randomised are six times more likely to have a positive effect [6]. In systematic reviews of animal models, the efficacy of treatments to reduce symptoms of multiple sclerosis was on average 1.4-fold higher [7] while the efficacy of a treatment for stroke (NXY-059) was on average 2.2-fold higher in the studies that did not report using masking [8]. These observations, along with concerns that much of the research published in the biomedical literature cannot be reproduced, has led many major stakeholders in the scientific community to call for improvements in the rigour and transparency of preclinical experiments over the last decade [9–13].

Masking is considered a gold standard to ensure the rigour and reliability of animal experiments [10,14]. The recently updated ARRIVE guidelines, endorsed by over 1,000 journals, now explicitly prompt researchers to report whether masking was used at the different stages of the experiment—during the allocation of animals to groups and the experimental interventions, during the conduct of the experiment (general care and welfare management), during the outcome assessment, and during the data analysis [15,16]. This follows a move observed in clinical trials, where trialists are now asked to describe explicitly who was masked and how, rather than relying on ambiguous terms such as single or double masking [17,18]. In publications reporting on animal research, an encouraging statistically significant improvement in the reporting of masking has been observed between 2009 and 2015 [19]; however, the prevalence is still very low. The Kilkenny study [20] found that only 14% of studies reported on the use of masking in an analysis of 271 randomly chosen articles. A more recent study, examining 2,671 published articles collected for systematic reviews of drug efficacy in eight disease models, found that reporting whether the outcome assessment was masked depended on the disease model (varying between 3% and 59%) with an overall average of 29.5% [21] and an automated analysis of over 50,000 animal research articles published in 2018 showed that the masking status was only reported in 12% of papers [22].

The reasons for such a low prevalence of reporting on masking are unclear. It could either reflect that researchers are using masking as a strategy to reduce bias in their experiments but do not disclose it in publications or that the low reporting prevalence represents the practice on the ground and that masking is predominantly not used in animal experiments. A self-reporting survey of researchers in Switzerland found that 27% (sample size = 530) had implemented masking at outcome assessment, but the survey authors noted that the self-reported rates were likely biased as they were considerably higher than the reporting rates found by

systematic review [23]. This observation was corroborated during user testing of the ARRIVE guidelines 2.0, where researchers often indicated their reluctance to disclose the masking status in their manuscript if masking had not been used in their experiments [15].

Here, looking at studies conducted within both academic and pharmaceutical settings, we have explored how masking is used in practice and discussed with scientists their awareness of the concept, the strategies they implemented, and the barriers they encountered. Common themes were identified as hindering scientists from implementing masking. We present the reality of masking in in vivo research, and, through a detailed exploration of case studies, we share the challenges and discuss the solutions to move forward towards better practice. A glossary (Box 1) explains the common terms used. This analysis will provide insight to researchers and anyone managing researchers looking to use masking in their studies, but also managers of animal facilities and institute leads looking to support their scientists.

### Box 1. Glossary

Definitions are placed in the context of animal research.

**Bias:** The over- or underestimation of the true effect of an intervention.

**Allocation concealment:** A workflow that conceals what experimental group each animal will go to until the intervention is applied.

**Allocation sequence:** Spreadsheet that randomly pairs each animal ID with one of the experimental groups.

**Confounding**: Confounding occurs when the design of the experiment does not eliminate plausible alternative explanation for an observed relationship.

**Experimental group**: A group of animals that receives the same intervention in an experiment.

**Intervention**: The process or action that is being investigated in the experiment. For example, a drug treatment or a genetic modification.

**Masking**: Masking (also known as blinding) is a methodological process where an animal's allocation to a specific experimental group is concealed from the people running the study, caring for the animals, or analysing the data.

**Outcome measure:** Any variable recorded during an experiment to assess the effects of an intervention.

**Treatment efficacy:** The effect of the intervention

## Methodology

Interviews were conducted with a variety of scientists who conduct in vivo experiments from both academic and commercial settings. The goal was to understand how masking was implemented in practice for different types of studies at each step of the experiment and what the challenges and barriers to masking were. In both types of settings, we use purposive sampling. We chose a broad cross section of research settings to account for the possibility of cultural differences and to ensure we had identified the various barriers that could arise, but the study was not designed to compare results between settings.

In the academic setting, NC3Rs Regional Programme Managers (ES, CB, and KAA) interviewed researchers at seven different universities in the United Kingdom. We selected one or two research groups per university/facility to cover a variety of research species and research types. Different researchers were interviewed for each of the 12 experiments; all researchers approached agreed to participate, but one interview was never scheduled due to the interviewer leaving the organisation (KAA). Ethical approval was obtained from the University of Leicester (ID 17587), which covered all interviews within the academic setting.

In the pharmaceutical setting, with the approval of AstraZeneca Council for Science & Animal Welfare (Enterprise-wide governance group for research involving animals), a statistician embedded with the community (NAK) reviewed all study types conducted at the time at two distinct facilities in Cambridge, UK. This led to an assessment of 18 study types, which was a complete coverage of experiment type conducted internally within the UK. The project was advertised through the local ethical review board. Scientists who conducted these studies volunteered and were assembled into representative groups to be interviewed. The group size and which studies they were interviewed about is captured in S2 Table.

All interviewers (NAK, ES, CB, and KAA) had lab experience, a PhD with formal scientific training, and between 7 and 14 years' experience working as, or supporting, in vivo researchers. In both settings, the interviewer and the researchers had an existing work relationship. We were aware that masking was not widely used in animal experiments but believed that the research community was open to using it and assumed there were practical barriers stopping them. The interviews were semi-structured (see S1 Table) and conducted as a discussion with the question order not considered. This format was designed to encourage open and honest reflection; both animal research and questionable research practices are sensitive topics that can hinder people sharing. For that reason, the interviews were not recorded and interviewers were explicit that any details allowing the identification of the research group would remain confidential. The agreement with the researchers did not include consent to share quotes nor detailed transcripts of the interview.

The brief to the interviewers was to explore how the experiments were conducted, assess whether, how, and when masking was used to conceal the intervention animals had received, or the experimental group they belonged to, from the researchers running the experiment. When masking had not been used, the interview also explored the reasons for not using it and whether it would be feasible to implement in future experiments.

The interviews specifically focused on four different stages of the experiment:

1. During the allocation and intervention: This includes all steps of the experiment where the animals are assigned to experimental groups and the steps when they receive the experimental intervention(s) [24].

2. During the conduct of the experiment: This element is focused on the housing and welfare management of the animals during the experiment and includes those caring for the animals, welfare checks, or administering welfare interventions (i.e., not the intervention defining the experimental groups) to ensure that all animals in the experiment are handled, monitored, and treated in the same way.

3. During the outcome assessment: This includes any step of the experiment where an outcome is measured, or a sample processed in preparation for a measurement.

4. During the analysis: This step refers to the data processing and statistical analysis.

Following the interviews, the data (study type, randomisation strategy, the masking level achieved, and the associated barriers) were tabulated and stripped of any identifying

**Table 1. Coding developed to classify barriers to masking following inductive iterative discussion of the data from the interviews.**

| Code | Definition | Example of the reasons provided by researchers during the interviews |
|---|---|---|
| **Culture constraint** | Following the way things are normally done. | Masking has not been used for that type of experiment in the past. Using masking is not the norm. |
| **Fear of errors** | Concern that masking will introduce errors and compromise the experiment. | Masking might lead to mistakes (e.g., dosing error), introducing welfare concerns and/or loss of data. Masking risks introducing errors. If the person entrusted with masking made a mistake, it would only be detected at the end of the experiment leading to a loss of several weeks of work. |
| **Ownership issues** | Concern that if others contribute to the experiment, it will lead to ownership or authorship issues and difficulties in attributing credit for the work conducted. | Working in partnership with others introduces challenges in defining ownership, which is critical to provide recognition and career progression. |
| **Belief in the value** | Belief that the use of masking would not add value to the experiment. | Masking not considered important, although the researcher was aware that funders require it. Other sources of variation were considered and the researcher did not think the risk from not masking was high enough. No evidence that masking is necessary for these sorts of experiments. |
| **Knowledge constraint** | Lack of knowledge of masking or an aspect of masking (e.g., how to use it in practice). | Masking had not been considered but the researcher was enthusiastic to use it in future experiments for steps where it can be easily implemented. |
| **Operational constraints** | A constraint arising from the experimental workflow implemented within the facility that is not specific to that experiment. | Cage cards in the animal facility reveal the intervention the animals have received. |
| **Practical constraint** | A constraint arising from the experimental workflow that is specific to that experiment. | Animals in different experimental groups have different phenotypes or visual differences. The interventions are visually different (e.g., different colour food in the cage). |
| **Resource constraint** | Limitation in the number of people who can work on the experiment. | Only one person conducting the entire experiment and no one else has the expertise to carry out procedure or analysis. No access to technical staff who were familiar with the details of the experiment. Not practical as other people in the lab are busy. |
| **Technological constraint** | A constraint arising from the setup of the technology used to collect information and data in the experimental workflow. | Group information visible in the IT system during data collection and submission. |
| **Welfare concern** | Concern that the use of masking will compromise the care and welfare of the animals in the experiment. | Need to know what each animal has received to ensure appropriate animal care. Masking might lead to dosing errors, which could impact on the animals' welfare. |

information. These data were discussed as a team (all authors excluding JCU) in an inductive iterative process to identify the themes and develop a coding strategy to classify the responses (Table 1). The removal of identifying information fully masked EJP and NPdS to the source of the data. Whereas NAK, CB, and EJS were partially masked as they could recognise the data from the interviews they conducted themselves. For each experiment, the coding assigned along with the data on which this assignment is based are presented in S2 and S3 Tables.

Limitations of the methodology: The interviews were conducted by different individuals across the dataset, and different interviewers might have asked different questions depending on their background knowledge. The sample is not random; rather, the study was designed to encompass a broad range of experiment types. The sample size is small but should be sufficient to identify the majority of the barriers researchers can come across [25]. This recommended sample size is based on a single application; therefore, our study design makes the assumptions that the barriers are equivalent across in vivo studies and not heavily dependent on experimental type. Data saturation was not assessed, which means that additional barriers might exist that our study did not detect. As explained above, the interviews were not recorded and we did not capture quotes. For most experiments, multiple participants were interviewed together and the experiments discussed collaboratively. The data collected are therefore associated with

the experiment, and not the participant, and cannot be used to explore the effect of career level on attitudes and perceived blockers. As data were collected from only one industrial setting, this also prevents comparisons between industry and academic settings. All data shared in this paper are based on notes taken by the interviewers, which limit the assessment of our interpretation. This study aimed to identify the themes around barriers to using masking in preclinical research, but not to quantify them or test a specific theory. As such, the coding for the themes identified in the interviews was developed as the coding took place (inductive coding). Although this approach is appropriate to identify frequent themes where no similar research has been conducted previously [26], it does have some limitations: There is risk of confirmation bias, and quantitative analysis cannot be performed on the resulting data. As this study has identified common themes and developed some coding, it can be used as a basis for future studies to develop a code set a priori and use it to test specific theories about masking in preclinical research. The recommendations presented in this paper have not been produced via a broad consultative process; they are based on the interview findings and subsequent analysis by the authors.

## Results

Interviews were conducted with 32 researchers at various career stages, including graduate scientists, PhD students, postdoctoral researchers and fellows, lecturers, and professors. Thirty different studies were explored across ten settings, which included both a pharmaceutical and several academic institutes (S2 and S3 Tables). A variety of experiments were discussed including safety, efficacy, model induction, and studies to explore mechanisms of action. These studies used mice, rats, rabbits, or zebrafish. As part of our analysis, we assessed the experiments to consider when or whether masking was relevant; for six of the experiments discussed, masking was not required at one or more stages of the experiment due to the experimental design implemented (Table 2). For three of these, masking was not relevant throughout the experiment as the objectives were to provide descriptive information, and, hence, the studies did not include a comparative group. Similarly, a dose escalation study examining the toxic effects of a compound had only one experimental group as the animals were exposed to a sequentially higher dose following a washout period. This design was implemented as the higher doses would only be used if the animals tolerated the lower doses. The design format is not ideal from an experimental design perspective as the dose is confounded by exposure period. This study, with only one experimental group and sequential exposure, cannot be masked during the allocation, care, nor outcome assessment. However, masking can be implemented at the analysis stage. Two of the studies relied on mendelian inheritance to allocate animals to experimental groups, and, therefore, masking could not be implemented during allocation. For most of the studies discussed (27/30, 90%), masking at one or more of the stages could be implemented to minimise potential bias. Two of the studies (6.7%) had implemented some form of masking at all four stages of the experiment. Only three studies (10%), did not use masking at any stage of the experiment when masking would have been appropriate. The remaining studies used masking at some stage during the experiment, either during the allocation and intervention, the conduct of the experiment, the outcome assessment, or the analysis of the data.

## Discussion

The quality of the strategies varied in the ability to minimise risk of bias being introduced and highlight that masking is not simply a "yes, it is masked" or "no, it is not." The strategies exist on a spectrum of effectiveness. Ideally, masking needs to conceal not only the intervention information (e.g., drug X) but also the experimental group membership (i.e., which

**Table 2. Experimental scenarios identified during the interviews for which masking was not applicable (as assessed by the authors).**

| Study type | Intervention | Number and type of experimental groups | Stage masking not applicable | Reasoning |
|---|---|---|---|---|
| Genotype-Phenotype studies | Genetic modification | 3 groups:<br>■ Wild type<br>■ Heterozygous<br>■ Homozygous | Allocation | Mendelian inheritance randomly allocates animals to an experimental group. |
| Neurovascular phenotypes in zebrafish larvae | Development of spontaneous trait based on genotype | 2 groups:<br>■ Spontaneous haemorrhage<br>■ No haemorrhage | Allocation | Mendelian inheritance randomly allocates animals to genotype, which leads to development of spontaneous trait. |
| Tolerability | Exposure to treatment | 1 group:<br>Treated group | Allocation<br>Conduct<br>Outcome assessment<br>Data analysis | Objective purely descriptive therefore no comparative group included in the data. |
| New model tumour growth studies | Subcutaneous injections of cells | 1 group:<br>Treated group | Allocation<br>Conduct<br>Outcome assessment<br>Data analysis | Objective purely descriptive therefore no comparative group included in the data. |
| Pharmacokinetic (PK) studies | Exposure to treatment | 1 group:<br>Treated group | Allocation<br>Conduct<br>Outcome assessment<br>Data analysis | Objective purely descriptive therefore no comparative group included in the data. |
| Escalating dose telemetry study | Exposure to treatment | 1 group:<br>Experimental group exposed to multiple treatments | Allocation<br>Conduct<br>Outcome assessment | No comparative group included in the data. However, masking possible during analysis when comparing data across the treatments. |

animals are grouped together in group 1, group 2, etc.) during allocation, conduct, and outcome assessment (Table 3). Experimental group knowledge may result in the researcher detecting patterns that can allow the researcher to start interpreting the response. This can then lead to animals being handled or data collected more consistently within each

**Table 3. Masking options during allocation, conduct, and outcome assessment.** We have classified the masking options as either low, moderate, or high based on the ability to minimise risk of bias.

| Extent of masking | What is masked? | Quality of strategy |
|---|---|---|
| Full | The intervention information is masked and each sample/animal/cage is individually coded, so researchers are not aware which ones belong to the same experimental group. Researchers do not have access to any information or clue revealing the intervention animals are receiving. | High |
| Partial | The intervention information is masked but group coding provides knowledge of experimental group membership. Researchers are aware which animals are grouped together and receive the same intervention, but not what the intervention is. | Low |
| | The intervention information is masked but the workflow can give some knowledge of experimental group membership. Each sample/animal/cage is individually coded but are processed in subclusters due to practical constraints (e.g., animals for the same experimental group are co-housed). | Moderate |
| | The intervention information is masked and each sample/animal/cage is individually coded but signs or characteristics exhibited by the animals can give some knowledge of the experimental group membership. | Moderate to low depending on how overt the signs are |

experimental group, which may introduce bias leading to misleading conclusions (either leading to apparent differences where in reality there are none, or by masking real differences). For some experiments, the animals in a cage received a common intervention because of concerns over coprophagia (consumption of faeces) and dosing errors. This means the researcher handling the animals knows that these animals in this cage were from the same experimental group. Where possible, it is recommended that researchers are masked to both intervention and experimental group membership. We will explore each stage, looking at the successes, exploring the pros and cons of strategies that were or could be implemented, discuss the barriers encountered that are unique to each stage, and then explore the common barriers.

## Masking during the allocation and intervention

The researcher's expectations can influence (consciously or subconsciously) the allocation of animals to an experimental group, the application of the intervention, and the handling of the animals during these processes. This can introduce systematic differences between experimental groups. Consequently, masking is necessary prior to the animals being allocated to an experimental group, during the allocation (allocation concealment) and during the application of intervention [24]. Examples of subconscious bias include the surgery being rushed in a sham animal, the handling of the animal being more cautious with a disease-induced animal, or the animals with lower tumour volumes being allocated to the control group. A formal randomisation procedure (e.g., the use of a computer-generated random allocation sequence), which ensures each animal has an equal probability of receiving a particular intervention, is also a critical tool to reduce systematic differences occurring between the experimental groups. However, randomisation alone is insufficient as knowledge of the group allocation may still influence the way animals are handled and treated before and during the intervention.

The interviews found that full masking (from both the intervention and the experimental group membership up to and during the intervention) was only implemented in one study. This example is presented as a case study in Box 2 as an example of best practice. However,

---

### Box 2. Case study: Masking during the allocation in an experiment with multiple interventions

Let us consider an experiment testing the effect of a pharmacological intervention on a surgical pain model. Animals first receive a surgical procedure—with some experimental groups receiving a procedure to induce the model while others receive a sham surgery. Then, they receive a drug treatment several weeks later—with different experimental groups receiving either a vehicle or a specific dose of the drug under investigation. Masking the investigators before and during these interventions ensures that there are no differences in the way animals from different experimental groups are handled before the surgery or during the drug injection and that the pre- and postsurgical care is identical in all groups (e.g., there might be a temptation to be stricter with the timing of the analgesic if the "real" surgery is expected to be more painful than the sham surgery). Knowing the group allocation might also lead to important differences in the intraoperative care and monitoring, or duration of surgery (and hence dose of anaesthesia), which could impact on the outcome of the experiment. In such an experiment, masking researchers to the type of surgery requires at least two people: a surgeon and an assistant. Having two people involved in the surgery is also advisable to ensure good aseptic technique, so masking would not require additional staff in many cases. Before the surgery, the assistant prepares the desired number of sealed envelopes (one for each animal)

---

containing one of the two types of surgery hidden on the inside. Then, the surgeon randomises the order of the envelopes and assigns one to each unique animal ID and writes it on the outside of the envelope. This way, at this stage, neither the surgeon nor the assistant knows what type of surgery is assigned to each animal ID. On the day of surgery, the animals are all prepared in the same way and the surgery can progress up until the step that differentiates the model-inducing surgery from the sham surgery (e.g., a tendon is manipulated and left intact in the sham surgery, but the tendon is cut in the model-inducing surgery). At this point, the assistant opens the envelope corresponding to the animal ID (without showing the animal ID to the surgeon) and tells the surgeon what procedure to perform. Masking the rest of the surgery is not practically feasible as it would require a change of surgeon and assistant. Masking the immediate post-op period would also require a second assistant who was not in the operating theatre during the surgery, which might not be feasible for most labs. Once the animal has recovered from surgery, the assistant brings it back to its home cage so that later on the surgeon cannot use cage location to identify which surgery an animal has received. As the assistant is now unmasked, they should not be involved in caring for the animals, but they can help the researcher remain masked with regard to the drug treatment. The assistant can generate the randomisation sequence to allocate animals from each type of surgery to the different drug doses and prepare coded syringes for each animal, so the researcher injects the animals without knowing the contents of the syringe. This prevents any subconscious bias in the way animals are handled and injected (e.g., if the researcher did not manage to inject the entire content of the syringe in one go, they might be tempted to stop if they know the syringe only contains saline, whereas they would make sure to inject the remainder if they know the syringe contains an active drug).

most experiments had implemented randomisation, which is considered an important complementary strategy to reduce systematic differences between experimental groups. For some studies, other strategies to reduce the risk of subconscious bias had been implemented, such as using support staff who are not invested in the research to allocate the animals to groups or complete the intervention (Table 4). Often researchers thought that randomisation alone was sufficient to protect against bias during this step of the experiment and had not realised that bias could arise during the allocation and intervention process.

A common issue identified during the interviews was that the point of intervention—and thus when masking should be implemented—can sometimes be difficult to define. Some animal experiments do not include researcher applied interventions (e.g., some phenotyping studies where the only difference between the experimental groups is the genotype) in which case this step is not relevant. In other experiments, there could be several interventions—for example, a surgical intervention and a drug treatment (see Box 2)—and implementing masking for each type of intervention requires careful planning.

Our discussions with researchers revealed that common barriers to using masking at this stage of the experiment were related to research culture and knowledge, as well as resource constraints. Researchers often reported that they either work alone and that other staff or students were not available or too busy with their own projects to help prepare coded treatments or administer them to the animals. Another barrier frequently mentioned was a reluctance to trust someone else with the responsibility of masking the main researcher. Entire experiments could be wasted if the group allocation information is lost and the researcher cannot be unmasked, or if a mistake (e.g., an animal received the wrong treatment) is only detected at

**Table 4. Example of strategies to implement masking during the allocation.**

| Type of intervention | Strategy to mask the intervention or the group allocation |
|---|---|
| Intervention is a drug injection—drug and vehicle have no discernible differences | An assistant/colleague/student loads and codes the syringes with the unique animal ID based on the allocation sequence, and the researcher therefore injects the animals without knowing the content of the syringes. |
| Intervention is a drug injection—drug and vehicle have visible differences | As above but opaque syringes are used. |
| Intervention is a drug injection—drug and vehicle have different viscosity | An assistant/colleague/student with no vested interest in the study loads the syringes and injects the animals based on the allocation sequence. |
| Intervention is a surgery—the surgeon needs to know what procedure to perform | See case study in Box 2. |
| Intervention is a tissue graft of different genotypes | Donor tissue is stored and genotyped at the end of the study. |
| Intervention is an inoculation with different cell lines | A different lab member, not involved in the in vivo experiment, grows the cell cultures and codes the cell vials before the inoculation. |
| Intervention has a safety concern and staff administering it need to wear protective equipment | Similar protective equipment is used to handle all animals in the study, including those receiving an inoffensive vehicle. The intervention is prepared by an assistant/colleagues/student who codes the intervention with the unique ID of the subject based on the allocation sequence. |
| Intervention is a diet | An assistant/colleague/student with no vested interest in the study aliquots the food into bags coded for each individual cage, and the researcher feeds the animals without knowing what the diet is nor which experimental group the cage belongs to. |

the end of the experiment. Some researchers had implemented strategies to reduce the risk of mistakes, including grouping the animals from the same experimental group within the same cages, or on the same cage rack. These strategies prevented full masking as they provided experimental group information that allows the researcher to potentially see patterns in the data or animals. Systematic differences between cages or racks may also confound the effect of the intervention and put the experiment at risk of additional bias [16]. To enable masking, teamwork is essential, and robust procedures have to be put in place to minimise the risk of mistakes and ensure that the group allocation information is kept safe and readily accessible if the information is needed to address welfare concerns. Ideally, the allocation sequence should be generated by someone not involved in the day-to-day care of the animals to enable consistent care across experimental groups. The Experimental Design Assistant [27] provides an option to generate an allocation sequence that can be emailed to a colleague or collaborator who can then be responsible for coding the intervention (e.g., diet or syringe) or administering the intervention (e.g., performing the surgeries). This sequence information then needs to be stored securely and be readily accessible. Ideally, this would be through local IT data management systems to remove the risk of losing the information and to provide ready access. Alternatively, it could be emailed to a senior manager (supporting vet, facility manager) and printed and stored in a common file in the facility. To address the fears and enable the researchers, locally applicable solutions need to be trialled and implemented within their operational systems (e.g., cage labelling or data systems that allow masking).

## Masking during the conduct of the experiment

Masking during the conduct of the experiment refers to the handling and welfare management of the animals at any point during the experiment [16]. Ensuring that caretakers and animal

care staff are not aware of the group allocation minimises the risk of performance bias that would occur if animals from different experimental groups received systematic differences in their care, or exposure to other factors [28]. It is widely accepted that the welfare of research animals should be optimised to deliver quality scientific research and to address our ethical obligations to refine under the humane use of animals framework. The implementation of strategies to optimise the welfare and monitoring of welfare through the life experience of the animals is consequently a specific legislative requirement in many countries (e.g., the Animals (Scientific Procedures) Act 1986 in the UK, the EU Directive 2010/63 or the 1966 federal Animal Welfare Act (AWA) in the United States). These welfare checks are critical to rapidly and effectively detect suffering so that appropriate action may be taken, which may include providing analgesia, reviewing husbandry (such as providing wet food), or euthanising the animal. Masking the animal care staff and those involved in the decision making is necessary to ensure that all animals in an experiment are handled, monitored, and treated in the same way. A welfare assessment protocol should be developed for each specific project to plan for welfare issues specific to the project, to develop a list of indicators with welfare intervention points and an action plan for these situations if they arise [29]. From an experimental design perspective, the ideal solution is that these welfare interventions should be applied regardless of experimental intervention the animals received. This approach will ensure consistency and remove potential bias being introduced into the experiment [29].

During the discussions with scientists, several scenarios of differential animal care were discussed, which unmasked the experiments. These could be grouped into two clusters: welfare needs that result in additional animal husbandry or unexpected/transient welfare needs that can lead to a welfare intervention. Consider the first group, which are studies that for some interventions can result in an additional welfare need requiring an additional animal husbandry step for all animals within that experimental group. For these, we argue that the solution to meet our ethical obligations and deliver robust science where masking is implemented is to care for the animals identically and provide that welfare intervention for all animals in the study. Consider the situation where an intervention leads to weight loss, which can be mitigated by an additional dietary supplement such as wet food. A provision of a dietary supplement introduces additional costs and time (cage cleaning, etc.), and thus it is tempting to only supply to the affected experimental group. This, however, is not recommended as the alternative diet will alter the behaviour of the animals within the affected experimental group and potentially the different diet composition could alter their physiology. Therefore, any differences observed could be a result of the intervention, the altered diet, or the unmasking of the study. Other examples of differential care discussed included a higher frequency of welfare checks in the more severely affected group or analgesic being given only to one of the experimental groups. In both these situations, the recommendation would be to provide equivalent care to all groups to minimise bias while meeting the welfare needs. Related are the scenarios that lead to differential care of the animals to maintain the welfare of the staff. In a scenario discussed during our interviews, hazardous microorganisms were used in the intervention for some of the experimental groups, and consequently additional steps were needed to protect staff. To mitigate the introduction of bias in these situations, the welfare management strategies should be applied to all animals regardless of which experimental group they sit within. In effect, treating the control animals as if they were as dangerous as the infected animals. These examples highlight the potential of differential care to introduce confounding into an experiment; ethically, it means the animals are wasted and this should be avoided even if it increases the costs associated with the experiments. The recommended approach of a common welfare strategy is a significant shift and will require exploration of the topic with all the stakeholders to agree the welfare assessment protocol going forward.

A number of interviews raised concerns around managing welfare needs while maintaining masking. The scientists raised examples where some interventions or model induction have a transient welfare issue that would be tolerated for certain experimental groups. For example, in oncology studies, the anticancer treatments being tested could induce moderate transient piloerection or hunched posture and reduced social interaction, which would be tolerated for a short period of time in the treated animals with additional monitoring, but the same welfare concern would not be tolerated in the control animals. Within these animal facilities, with a legal and ethical culture of care towards individual animals [30], the same welfare issues seen with a control animal where there is no known reason for the effect is likely to require a humane termination, while with the treated animal if the effect is expected, additional monitoring and husbandry support might be provided rather than humane termination. The timing of the welfare observations and understanding of expected versus unexpected events in relation to different interventions is critical to the welfare decision. This is an interpretation of the ethical obligations, and the management can be the same for many situations for both animals. The welfare management plan is an interaction of the ethical legal framework, the experimental detail, the severity of the welfare issue, and the culture of the institute. There are, therefore, some situations, where the ideal experimental solution cannot be implemented. In these situations, a harm–benefit analysis is needed to explore the ethical challenge of the risk of bias over the risk of harm for individual animals. Where it is found that unmasking will (or may) be required, a clear plan should be put in place in advance of the experiment starting, to determine which individuals are to be unmasked, and at what stage this may be required. In discussing the plan, conflicts may be revealed between different parties, but it is important that decisions are reached so all parties know how to proceed; this will help to keep any bias to a minimum. The PREPARE guidelines can help with this as part of the suggested dialogue between the scientists and the animal facility [31]. It is important to acknowledge the biased care in these studies and whether the evidence generated by the experiment will be reliable enough.

Regardless of the experimental detail, technology and operational constraints were a common barrier to masking during the conduct of the experiment. This typically impacted both masking during conduct and masking during outcome assessment for in vivo measurements. For some studies, the operational implementation often requires experimental details to be readily accessible to manage welfare concerns and minimise errors. For example, intervention information is often found on the cage cards, input pages for data collection, or associated documentation for those experiments. Therefore, with the visibility of the intervention information, the possible strategies to deliver masking within these systems are inefficient as they require the scientist to operate outside of the data management systems and can be resource costly as they require a second operator who interacts with the system while the first interacts with the animals. Furthermore, if they require additional steps to transfer data or information, they are error prone, which introduces ethical concerns over wasting animals. Consequently, the technological and organisational management systems can limit the implementation of masking, and, in these situations, masking is not an individual scientist issue but a facility-level issue. Developers of the IT systems and the integration of these systems by animal facilities management into the daily operations need to refine the information flow through their systems such that masking is enabled, while ensuring that the welfare information is readily accessible (for example: coded IDs combined with alternative pages with detail information on intervention received). In situations where welfare issues arise, the detail of the intervention can be revealed to those critical in the decision-making process (e.g., the supporting veterinarian). It is critical that these issues are considered during the setup of facilities, development of new systems, acquisition of new equipment, and establishment of new experiments. By

building in the functionality, this will enable scientists to follow best practice and normalise the expectation.

Regardless of the operational systems, for some experiments, visual differences in the animals can indicate the intervention received. The use of support staff, who are less invested in the pressures of desiring or anticipating a particular experimental outcome (for example, compared to a Principal Scientist for whom future plans may depend on an experimental result) would mitigate against the introduction of bias in these scenarios, particularly if details around intervention are hidden and welfare management strategies are consistently applied across the animal set. However, there may be cases in which these experimental interventions themselves make the allocation clear, limiting the application of masking at the stages of the experiment that involve working with the animals.

## Masking during the outcome assessment

Masking during the outcome assessment refers to the researcher being masked during the step (s) of the experiment where an outcome is measured. This is critical to reduce potential bias where differences can be introduced by systematic differences between the experimental groups in the way in which the outcome is assessed [28]. Examples of outcome measures include the observation of a behaviour, cell counting, image capture and analysis (quantification), etc. For many experiments, the point of measurement is not a clearly defined stage, such as those that have multiple processing steps before any data are obtained. For example, ex vivo measurements of samples using techniques such as flow cytometry, RT-PCR, or single-cell proteomics all involve processing steps in which knowledge of the experimental group could influence the handling of the sample, albeit unintentionally, leading to the introduction of bias in the data ultimately obtained. This was demonstrated in the "memory of water" study, which was initially thought to demonstrate the scientific basis for homeopathy. The initial paper reported evidence that basophils can be activated to produce an immune response by exposure to solutions of antibodies that had been diluted so far that they contained none of the original antibodies [32]. With the implementation of masking during cell counting, this consistently observed treatment effect was no longer observed [33]. This is an example of a quantitative measure that demonstrates the potential impact of prebeliefs on the measured outcome. This challenges the belief that only subjective measures need masking and the interaction will arise through subtle interaction points that exist between the assessor and the data obtained. For example, when measuring a tumour volume with callipers, there is a decision of where to place the device, in a study counting neurons in a brain slice the researcher must select a suitable area to count, in a behavioural assay there are frequently manual handling steps where the expectations of the operator can influence the outcome.

Our interviews with researchers identified a variety of masking strategies with varying degrees of robustness were used when collecting measurements in vivo. A common strategy when researchers knew the intervention the animals had received was the use of support staff or students who collected/conducted the measurements with the idea that these individuals were less inclined towards achieving one experimental outcome or another (for example, in comparison to the Principal Scientist whose research plans may be based upon a particular hypothesis), and, hence, this would reduce the unintentional observer bias effect, particularly in high-throughput environments where large and multiple studies are being conducted. However, while the support staff may be less inclined towards a particular experimental outcome, they will likely be aware of the goals of the researcher, which has the potential to introduce bias if the support staff are aware of the intervention received. Power dynamics, or a want to please (whether knowingly or otherwise), could also be at play, particularly when the support staff is

junior and part of the same department, decreasing the effectiveness of this strategy to reduce bias.

Technology can reduce the risk of observer bias for some assays, for example, where measurements are automated. Examples seen included an automated calliper system for measuring tumour volume or the automated testing system with the rotarod, which detects the animals falling off the via a magnetic tray. However, the handling of the animal into the system could be influenced by knowledge of the intervention, and, ideally, the observer would be masked to this information. For some researchers, the setup of information to manage welfare concerns and minimise errors meant that intervention information was placed on the cage cards, on input pages for data collection, and on associated documentation (technology/operational constraint to masking; see Masking during the conduct of the experiment). This data flow and organisation limits the implementation of masking, but solutions had been found in some of the examples discussed. For example, The Irwin assay, a neurobehavioural assessment [34], employed support staff to pass the animals with identifying details hidden to allow assessment without bias. This strategy allowed the researcher to be masked despite technological and paperwork systems showing the intervention received, but it required an additional person. To implement this approach therefore requires sufficient resources to be available and collaborative working to be acceptable.

When measurements were collected ex vivo, successful masking depended on how the samples were labelled and how accessible the allocation information was. As with in vivo studies, several experiments were using independent researchers (external contract research organisations, support staff, or students) to process the samples, which reduces the risk of bias. The more independent the researchers, the more likely the risk will be reduced. In high-throughput studies, labelling samples with only the sample ID would reduce the risk of bias as the researcher is unlikely to remember what intervention the animal had received. However, it would be better practice to use a masked sample ID to remove the risk completely. Our interviews found that samples were often labelled with both a masked sample ID and a group ID. This achieves the goal of masking to intervention but not which samples are grouped together and further refinement to only include masked sample ID would minimise the risk of bias further (see Table 3). A formal review of masking (at the lab or facility level) can identify opportunities for refinement to the study implementation. This was seen for the flow cytometry studies where the interview found that masking was not implemented for data processing as the researchers required knowledge of the control group to set the gating settings. This workflow has since been refined and a standard pooled sample is now used for the establishment of the gating parameters.

The assessment of histological slides is a subjective process; however, many pathologists are resistant to masking samples, arguing that it will reduce sensitivity as they rely on "inductive reasoning" [35]. Inductive reasoning is necessary as histopathology changes can represent part of a continuum or variation from "normal" background findings. This, along with experimental variation in sample processing, e.g., staining and sectioning, results in a need to understand the normal pathology. With this knowledge, pathologists can then identify the thresholds for determining abnormal pathology as well as assigning severity scores when assessing for unknown intervention effects. In the context of toxicology studies, it has been argued that observer bias is conservative with regard to safety and is biased towards identification of a toxicological hazard [36]. A compelling example for the need to mask histology samples comes from two studies assessing the same set of slides from the National Cancer Institute [37]. In one assessment, which made no mention of masking, the researchers found that the intervention had significant carcinogenicity and toxicity effects such that they identified 23 different tumour types to be intervention related. [38]. An independent study, where the assessments

were masked to the intervention, found no tumours to be intervention related. An international group of pathologists, when considering reporting guidelines for pathology data [35], recommended that when studying a new system, masking can be achieved by completing an initial assessment of all tissues with full knowledge of the experimental details to determine the thresholds to ensure all findings are detected, and then use a subsequent masked review to confirm subtle or borderline differences in selected tissues. Where possible, the integration of a Pathology Peer Review during step 1 can improve the accuracy of interpretation by providing a step to ensure consensus of thresholds that are then applied [39]. This multiple-step approach is labour intensive but protects against observer bias. Furthermore, the process implicitly indicates that inductive reasoning is only necessary in situations where the outcome of intervention is unknown. With targeted research using established models, when assessing defined abnormalities, masking can be implemented from first assessment [37].

## Masking during the data analysis

Masking during data analysis refers to the analyst being masked during the step(s) of the experiment where the resulting data from an experiment are processed, and statistical analysis conducted. Masked analysis ensures that all analytical decisions have been completed before the relevant results are revealed to the researcher. Masking is necessary as data analysis has many points where subjective decisions are made, for example, how missing values will be handled, whether to transform data, whether to include covariates, decisions on outliers, and statistical test selection. There is a real risk of significance chasing or p-hacking [40], where the analysis pipeline is amended to achieve the goal of finding a statistically significant difference between the experimental groups [41]. This potential interaction of the researcher with the data leading to bias has been used to explain an observation in particle physics where new values are closer than expected to published values given the known standard errors of measurement [42]. Another option to reduce the risk of bias is to preregister the analysis plan in detail, making it clear how data will be handled and analysed before it is collected. However, this may limit the scope of analysis, prevent a change of approaches to use more appropriate statistical strategies for unexpected anomalies in the data, or become overly complex when trying to define the decision-making process that led to the analysis choices made. Amending the preregistered analysis plan is acceptable if the details are disclosed [43]. Even if analysis is considered exploratory, masking is important to minimise the potential of the analyst introducing bias into the conclusions.

Numerous masking strategies have been discussed in the literature and they perturb either the data values and/or the experimental group labels (Table 5) [42]. The technique must obscure the data with meaning while showing enough of the data to allow for decisions to be made about subsequent analysis. Provided the analysis plan is then followed, p-hacking is avoided as the decisions are not being driven by whether the statistical outcome is significant or not. These strategies all require at least two parties: a data manager who masks the data and an analyst who designs the analyses. The strategies vary in their effectiveness at minimising the risks, the ease of application, and applicability. Among those interviewed, masking during data analysis was infrequently implemented, although examples included the use of an independent analyst and randomly coding the experimental groups. These two strategies are easiest to implement, are applicable to a wide range of studies, and, if combined, deliver a high quality of masking and confidence that the risk of bias being introduced by the analyst is minimised. Other strategies are being actively explored but are not universally applicable and are more difficult to apply so are not discussed further here. Overall, our study suggests that masking during analysis is not embedded in our research culture and has not been deemed critical, so there

**Table 5. Example of strategies to implement masking during data analysis.** We have classified the masking options as either low, moderate, or high based on the ability to minimise risk of bias. Shown in bold is the recommended high-quality strategy that is readily implementable across different experiment types.

| Strategy | Additional detail | Strengths and weaknesses identified by the authors or the literature | Quality of strategy |
|---|---|---|---|
| Independent analyst | The analyst could vary in their independence. Examples vary from a student in the same lab to an independent statistician. | Weakness:<br>• While not emotionally invested in the outcome, the analyst is still fully aware of the significance of 0.05 and the needs of the researcher.<br>• Power dynamics could be at play particularly when the analyst is junior and part of the same department, which will decrease the effectiveness of the strategy to reduce bias | Low-moderate |
| Randomly code the experimental groups | For example: In a study with four experimental groups (e.g., control, low, medium, high dose), the groups could be recorded randomly to group W, group X, group Y, and group Z. | Weakness: Can still see the experimental group differences.<br>Strength: Simple to implement. | Moderate |
| **Independent analyst and randomly code the experimental groups** | | Strength: Effective and readily implementable across different experiment types. | High |
| Adding noise | Add noise (from an appropriate statistical distribution) to the outcome measure to hide the true relationship between intervention and outcome measure. | Weakness:<br>• Precise amount being added is important to be effective and not alter the properties of the outcome measure.<br>• Requires statistical and computational sophistication. | High |
| Decoy data analysis | Analysts works with multiple data sets (e.g., 6), one of which is the original. | Weakness:<br>• Many of the masking strategies rely on knowledge of what matters in the data, aspects of the data might be unexpected, and, therefore, implementation might eliminate existing effects, induce effects, or change directions of effects [45].<br>• This strategy does increase the analysis time.<br>• Requires computational sophistication | High |
| Shuffle the key variables | For example, in a correlation analysis, a scientist shuffled the outcome measure but kept the relationship between independent variables of interest [46]. | Weakness: Requires computational sophistication. | High |
| Adding cell bias to equalise the means | Hide experimental group differences by adding the same hidden number to all observations within an experimental group, which leaves the distribution intact but obscures the differences [42]. | Weakness: Requires computational sophistication. | High |

is limited awareness of the topic and how it could be implemented effectively. Masking during analysis does require more resources as it needs two people working together (data manager and analyst) and additional time or expertise to implement. However, this can be achieved with minor changes to the process of analysis, so this should not be a barrier to it being implemented as standard.

Best practice for data analysis is to use software (for example, R; [44]) that enables reproducible data analysis, where the data and analysis script can then be made available. With these tools, the steps undertaken are inherently documented and can be reproduced. Having systems in place that support reproducible data analysis also supports masking as the analysis can be undertaken with the masked data, and with an additional line or two of code can be rerun with the correct classification or original data. Alternatively, scientists can continue to use point and click software (e.g., GraphPad, InVivoStat, Minitab) but would need to record every step and decision during the analysis on the masked data prior to unmasking.

As an example, consider an experiment where we have four experimental groups (control, drug A, drug B, and drug C) and we are interested in assessing for a treatment effect on the mean of the outcome measure. This analysis would typically be conducted with a one-way ANOVA with post hoc testing to compare the groups of interest. During the analysis, the analyst will make decisions on how to manage missing values, whether to transform the data,

whether to exclude outliers, whether to assume equal variance, and whether to assume normality. To minimise the multiple testing burden, planned comparisons would be conducted, only comparing data from each drug group to the control group. To enable masking, the analyst would randomly code the experimental groups while making the data analysis decisions. However, to enable the planned comparisons, the analysis must be rerun following unmasking.

## Barriers to masking that are common to multiple experiment stages

During the interviews, many general thoughts on the topic of masking were expressed that can help us as a community understand why it is not implemented as standard. These thoughts are captured in Table 1 along with the associated classification of these barriers. This section will explore these common themes and reflect on how those supporting the scientific community can enable scientists to implement masking as standard practice in animal research.

For some researchers, it was apparent that they were unaware of masking as a concept (knowledge constraint). This highlights that, while masking is considered a standard attribute of experimental practice, it is not the cultural norm within the preclinical research community. Several researchers were aware of masking as a concept but were resistant to implementing it due to beliefs that question the value of masking (belief in the value). For example, the scientist would respond to questions on masking with pushback, arguing that there was no evidence that it really mattered for their sort of experiment. Some of this suggests that researchers have a knowledge gap of the evidence that exists on the value of masking but could also represent cultural resistance to change as it is not the norm and current practices are seen as "good enough." The observed resistance could also arise from a knowledge gap of pragmatic solutions; when it seems too challenging to achieve, then implementing masking does not seem worth the effort, or worth the risk if researchers are concerned the process might compromise their experiment.

In fact, fear of errors was a common theme of resistance to masking. If mistakes are made, the costs are high both in terms of project progression but also ethically for wasted animal lives. The chances of errors increase with the number of people involved and number of masking/unmasking steps necessary. This was expressed in terms of dosing errors, management of welfare, and running complex experiments. This is particularly an issue when an easily implementable strategy to achieve masking is to use support staff or work with a second researcher, or when using operational and IT systems that were not designed to facilitate masking. Welfare monitoring and IT systems within organisations can be great tools to reduce errors but can also be an operational constraint to masking. This means the institutes and facilities themselves need to enable researchers by altering these systems to allow for masking while still meeting ethical and welfare expectations.

In addition, within a facility, we need to consider the resources available to the researchers. Many of the strategies discussed in the preceding sections require a second person, for example, a data manager or a member of support staff for subjective assessment of an animal. The availability of skilled technical staff within academic facilities was a common hindrance to enacting masking in the case studies, making researchers reliant on collaborative working. This arguably only shifts the resource constraint from the facility to the individual research group and can also introduce ownership issues over work, which can in itself be a significant barrier, both in academic and commercial research facilities.

While analysing interview data, we identified a number of experiments that had been implemented in such a way that they introduced potential confounding effects that could lead to systematic differences between the experimental groups. For example, in one study, the researchers allocated animals in poorer condition to the control group to avoid risk of animal

loss during the intervention. In another, larvae from different experimental groups were placed in different multiwell plates to reduce the risk of human error. The researchers had implemented these designs with a desire to minimise mistakes, variation, or welfare concerns. These issues reduce the ability to isolate the treatment effect, and the prevalence represents a knowledge constraint around experimental design.

Overcoming each of these barriers will require a multipronged approach including additional training, changing the cultural norms, and increasing both support and advice/guidance for researchers. Insufficient formal experimental design training has been previously noted, leading to a call to integrate this training into our education system [47–49]. However, we also need to provide on-the-job training to address this knowledge gap for scientists who are conducting or designing experiments and have completed their formal education. We recommend institutions provide experimental design training and consider integrating this into continual professional development (CPD). Such training should include the need to mask and the strategies for achieving this. The British Pharmacological Society has developed a curriculum for the use of research animals, which includes experimental design issues and have collated resources for educators to use [50]. Addressing the lack of availability of support staff within academic facilities may require action at the facility level and/or from managers of individual research groups in considering how to support collaborative working to enable their research staff to implement masking.

To drive improvement of current practices, we need to challenge underlying beliefs questioning the value of masking, the fears that are preventing the engagement with potential solutions, and raise expectations so that masking is the norm and is acknowledged as important for quality science. The inclusion of masking within planning and reporting guidelines [10,15,51,52,53] and the request to report on all four stages of an experiment is part of the journey to normalise this expectation. The engagement of journals, funders, and professional bodies with these quality markers will assist in the culture shift that masking should be a standard practice.

This manuscript has shared some practices and ideas on how masking can be implemented. These, along with the obstacles and perceived barriers to further implementation, were revealed simply through discussions with researchers. This suggests that an internal review within facilities, potentially driven by ethical review boards, is a critical step in assessing the current situation as well as beginning to address any shortfalls. This may require the development of a policy highlighting masking as an expected part of good experimental practice within the institution, supporting allocation of resources to drive the activity to explore local changes needed to enable this practically.

## Conclusions

Masking is a gold standard methodological process that has been demonstrated to reduce the risk of researchers unknowingly influencing the experimental outcome. Within an individual experiment, multiple approaches can be used, whether it is at all stages of the experiment or only a section. The methodology implemented will sit within a spectrum of its ability to minimise the risk. Researchers have to make pragmatic choices, taking into consideration welfare, practical constraints, regulation, and costs. Aiming for a perfect solution for all experiments is unattainable but if, as a community, we embrace the concept that masking is on a spectrum and value movement along that spectrum, we enable researchers to make small incremental changes that reduce the risk of bias. Experiments that are not masked are still valuable to publish, but this information should be disclosed to enable readers to assess how this affects the reliability of the results.

Implementing masking is not just an individual researcher's responsibility as the research represents the institute where it is conducted; it depends on the IT, operational systems, and resources available within an individual's institute. This finding is in keeping with the recent opinion article promoting the need to embrace ethnographic methods, which look at people in their cultural setting, as a critical step to enable scientists to embrace reproducibility reform [54]. This research provides a platform for further qualitative research to explore these issues in more depth, for example, to explore whether the prevalence of specific barriers relates to the career stage, environmental setting (e.g., academic versus industry), or biomedical research field. This manuscript provides practical advice both to the individuals and the institute management to enable masking; from the need to provide training, the need to optimise operational works and the need to generate a collaborative working environment where scientists can implement a masked workflow. The research presented here shows that masking can be used in a wide variety of settings across multiple diverse types of study.

## Supporting information

**S1 Table. Interview guide questions.** The term blinding is used, as this was the terminology used during the interviews.
(DOCX)

**S2 Table. Results and analysis of the interviews conducted at a pharmaceutical company during 2018.** The table presents the data collected during the interview, including the randomisation and masking status for each step of the experiment, and the barriers identified for each study. In bold, we indicate the coding assigned to each barrier during our analysis. These in-house rodent experiments, conducted within the UK, assessed either the efficacy, toxicity, or mechanism of action for a pharmaceutical intervention (e.g., drug treatment). As the experiment type was conducted by multiple people, the answers were obtained from group interviews and represents the typical setup within the company. As some issues are common across experiments and arising from the framework within which the studies are conducted, these have been grouped together for conciseness. The codes assigned to a particular study are not a fixed attribute of that study type but reflect the way these experiments were conducted. There were 8 interview groups where the number of researchers in each group varied from 1 to 3. No researcher sat on multiple groups. Which group discussed which studies are captured in the study type column.
(DOCX)

**S3 Table. Results and analysis of the 12 interviews held at seven different UK universities in 2018–2019.** This table presents the data collected during the interview, including the randomisation and masking status for each step of the experiment, and the barriers identified for each study. In bold, we indicate the coding assigned to each barrier during our analysis. The coding assigned to a particular study are not fixed attributes of that study type but reflect the way these experiments were conducted.
(DOCX)

## Acknowledgments

We thank Dr Kamar Ameen-Ali (KAA, former NC3Rs Regional Programme Manager) who collected some of the interview data from academic institutes. We are also grateful to the scientists who shared their practice and thoughts allowing the exploration of masking in practice.

## Author Contributions

**Conceptualization:** Natasha A. Karp.

**Formal analysis:** Natasha A. Karp, Nathalie Percie du Sert.

**Investigation:** Natasha A. Karp, Emma J. Stringer, Chris Barkus.

**Methodology:** Natasha A. Karp, Nathalie Percie du Sert.

**Project administration:** Natasha A. Karp, Nathalie Percie du Sert.

**Writing – original draft:** Natasha A. Karp, Nathalie Percie du Sert.

**Writing – review & editing:** Natasha A. Karp, Esther J. Pearl, Emma J. Stringer, Chris Barkus, Jane Coates Ulrichsen, Nathalie Percie du Sert.

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
