## [Editor Report · Decision Letter 0]

24 Jan 2022

Dear Natasha, 

Thank you for submitting your manuscript entitled "The reality of blinding during in vivo research: an exploration of engagement, barriers, and strategies to move forward" for consideration as a Meta-Research Article by PLOS Biology.

I refer you to my previous email to you and Esther for the details of this process; I apologise for these extra hoops that our system will make you jump through. I note that Esther has confirmed that any Article Processing Charges will be covered by NC3Rs in the event of publication.

Your manuscript has now been evaluated by the PLOS Biology editorial staff, and I'm writing to let you know that we would like to send your submission out for external peer review.

However, before we can send your manuscript to reviewers, we need you to complete your submission by providing the metadata that is required for full assessment [IMPORTANT - this is a formality in your case, as MOST of the metadata was already provided as part of your initial submission. HOWEVER, I did have to make up a couple of answers, so it would be good if you could check that you're happy with the metadata. Regardless, you will need to re-submit]. To this end, please login to Editorial Manager where you will find the paper in the 'Submissions Needing Revisions' folder on your homepage. Please click 'Revise Submission' from the Action Links and complete all additional questions in the submission questionnaire.

Once your full submission is complete, your paper will undergo a series of checks in preparation for peer review. Once your manuscript has passed the checks it will be sent out for review. To provide the metadata for your submission, please Login to Editorial Manager (https://www.editorialmanager.com/pbiology) within two working days, i.e. by Jan 26 2022 11:59PM.

If your manuscript has been previously reviewed at another journal, PLOS Biology is willing to work with those reviews in order to avoid re-starting the process. Submission of the previous reviews is entirely optional and our ability to use them effectively will depend on the willingness of the previous journal to confirm the content of the reports and share the reviewer identities. Please note that we reserve the right to invite additional reviewers if we consider that additional/independent reviewers are needed, although we aim to avoid this as far as possible. In our experience, working with previous reviews does save time. 

If you would like to send previous reviewer reports to us, please email me at rroberts@plos.org to let me know, including the name of the previous journal and the manuscript ID the study was given, as well as attaching a point-by-point response to reviewers that details how you have or plan to address the reviewers' concerns. 

Given the disruptions resulting from the ongoing COVID-19 pandemic, please expect some delays in the editorial process. We apologise in advance for any inconvenience caused and will do our best to minimize impact as far as possible.

Kind regards,

Roli

Roland Roberts

Senior Editor

PLOS Biology

rroberts@plos.org

---

## [Decision Letter · Decision Letter 1]

29 Mar 2022

Dear Natasha,

Thank you for submitting your manuscript "The reality of blinding during in vivo research: an exploration of engagement, barriers, and strategies to move forward" for consideration as a Meta-Research Article at PLOS Biology. Your manuscript has been evaluated by the PLOS Biology editors, an Academic Editor with relevant expertise, and by three independent reviewers.

You'll see that all three reviewers are positive about your study, but two of them (reviewers #1 and #3) raise a number of overlapping concerns about the reporting, clarity of the methodology, data availability and relationship to ARRIVE. All concerns will need to be addressed for further consideration.

In light of the reviews (below), we will not be able to accept the current version of the manuscript, but we would welcome re-submission of a much-revised version that takes into account the reviewers' comments. We cannot make any decision about publication until we have seen the revised manuscript and your response to the reviewers' comments. Your revised manuscript is also likely to be sent for further evaluation by the reviewers.

We expect to receive your revised manuscript within 2 months. 

**IMPORTANT - SUBMITTING YOUR REVISION**

*Re-submission Checklist*

*Published Peer Review*

*PLOS Data Policy*

*Blot and Gel Data Policy*

Sincerely,

Roli

Roland Roberts

Senior Editor

PLOS Biology

rroberts@plos.org

REVIEWERS' COMMENTS:

Reviewer #1:

[identifies herself as Nicole C. Nelson]

This manuscript reports on an interview-based qualitative study of researchers' self-reported practices and perceptions of blinding/masking in preclinical biomedical research. The data presented in this study is quite unique since there are at present very few qualitative studies of research practice and reporting behaviors, and therefore in my opinion quite valuable. But, the reporting (in particular the reporting of the methodology) is very incomplete, which makes it difficult to judge the quality of the interpretive work being done here. 

1. The authors should include a qualitative research reporting checklist with their manuscript—either SRQR or COREQ would be appropriate, but I think COREQ would be the better choice here since this is an interview-based study. Many of the problems noted below would be addressed by working through this checklist.

2. The reporting of the methodology is very incomplete. The following elements at minimum should be included:

a. The sampling strategy used to identify participants. The manuscript currently states that this sampling strategy is nonrandom, which is fine, but the specific type of nonrandom sampling strategy needs to be stated (e.g. is it just a convenience sample, or is it a purposive sample, snowball sample, etc.). It would also be helpful to include as much information as is known about the population that declined to participate and how their demographic characteristics differ from participants.

b. The number of interviewees and their demographic characteristics. Right now we only know how many types of studies the interviewees described, but we don't know if one interviewee described multiple studies; if the same study types were described by multiple interviewees; if the same study types were described by participants in both academic and pharmaceutical settings, etc. The lack of this information makes it extremely difficult to differentiate between concerns that are due to the type of study, the knowledge/perceptions of the researcher, and the institutional setting.

c. Whether an interview guide was used, and if the interviews were structured/semi-structured/unstructured. Right now we only know that interviews focused on blinding at four different stages of the research process, but there is no other information on steps the research team took to ensure that the same types of questions were asked across sites; whether there are important considerations about question order that were taken into account in the interview guide, etc. Information should be reported on which study team member interviewed which participant, and how the background knowledge of the study team member might have impacted the questions asked. This is particularly important if an interview guide was not used. 

d. Who conducted the qualitative data analysis (QDA), and what analytical approach was used. The interpretive approach appears to be more of a content analysis/thematic coding rather than a grounded theory analysis based on the information provided in Table 5. In addition, we need to know whether this approach was inductive or followed a pre-specified code book, whether an iterative approach was used, whether constant comparison was employed, etc.

e. What measures and thresholds of intra-rater reliability were used, and if multiple study team members were involved in the QDA process, what measures of inter-rater reliability were used. Were there any thematic codes on which the research team failed to achieve an acceptable level of IRR, and if so were these dropped from the study? A full description of the codes/coding tree should also be provided.

f. What measures of data saturation were used. In particular, the study should report whether termination of data collection was determined by interviewee characteristics or by saturation in the QDA process. 

g. Major themes and minor themes should be reported so that the reader can get some sense of how prevalent different concerns about blinding were, and whether the prevalence of concerns relates to career stage, biomedical research field, academic vs. industry location, local cultures around animal welfare, etc. Positions not taken in the data should also be reported, and analysis of positions not taken should be considered as it can be very fruitful in cases such as this one where there are potentially sensitive topics that may arise (e.g. questionable research practices that could be interpreted as misconduct, animal welfare concerns, power dynamics between junior and senior lab members). See Adele Clarke's _Situational Analysis_ for an excellent description of how to analyze positions not taken. 

3. The manuscript at present contains no representative quotes from the actual interviews. This is quite unusual for a qualitative study and means that the reader has not been provided any opportunities to compare the original data against the interpretations made by the study teams. For example, in Table 5, rather than paraphrasing participants' concerns, the table should put representative quotes next to their respective coding classifications. Ideally the demographic characteristics of the interviewees should be reported along with both quotations and paraphrased findings throughout, for the reasons described in 2b above.

4. When the authors say that the data are fully available without restriction, do they mean the full interview transcripts? If so, where are these made available?

5. The authors might consider using "masking" throughout as an alternative to "blinding," given that the use of blinding in this context promotes an ableist stereotype that being blind can be equated with lacking knowledge.

I suspect another round of review will be needed here, because it is difficult to assess the quality of the interpretive work performed and the solutions proposed without the methodological detail outlined above. For example, I am inclined to agree with the authors' argument that masking by assigning tasks to less knowledgeable/invested lab members is an insufficient strategy, but it would be useful to know if the study included interviews with any technicians/grad students/junior lab members so that we know if there are discrepancies between PIs' perceptions of these employees' knowledge versus their employees' self-reported awareness of the expected outcomes. In other words, while I am inclined to agree with this argument based on my own prior experience, it's difficult to know at present whether this is an argument that is supported by the data collected or whether it's a hypothesis that should be investigated in future studies. Once the manuscript includes more detail about the participants included in the study, then the presentation of these arguments can be more easily assessed.

Finally, one small note—one study that the authors should be aware of is Reichlin et al 2016 (10.1371/journal.pone.0165999), which addresses the question of whether the lack of reporting on blinding in the literature correlates to lack of blinding in practice. Contrary to what the authors argue in their introduction, Reichlin et al's data suggests that blinding, randomization, and sample size calculations are all performed more often in practice than they are reported in publications (although it is worth noting that their survey is based on self-reported data).

Reviewer #2:

The manuscript by Karp et al is an important and timely contribution to improving the quality of in vivo research, addressing application of blinding in different stages of experiment. I think it is very well written, based on the experience of authors and backed up by interviews and case studies. As the principle applies across species and study settings, I would only suggest to make (re-phrase) the statements to reference 1 and 2 in the introduction more general. Especially the work of Rosentahl and Fode (1) is so classic example which goes well beyond rats - ie it does not matter if you are studying worms, rats or humans, the recordings are biased without proper blinding.

Reviewer #3:

This manuscript reports the results from a qualitative study focusing on scientists' knowledge and perceptions on blinding of experiments to improve their reliability and reproducibility. 

Before I proceed with my comments, a declaration of interests: I'm familiar with the work of some of the authors, most of which I consider of high quality, which might also bias my appraisal. On the other hand, I have been critical - often publicly - of the organization for which some of them work in regard to the overreach given to the ARRIVE (granted, not always by said organisation itself) as a tool for the planning of experiments, for which these guidelines were not originally designed. 

As regards the manuscript itself, I find it has merits, it is well written, and I appreciate the depth of analysis of the issues at stake. I do have, however, to make the following remarks:

1. Although the research goal is laudable, the design has limitations, one of which being, ironically, that the study cannot be blinded at any stage, which is not acknowledged by the authors. And it should be, particularly given the topic covered.

2. It is not clear from neither the Methodology section nor the Contributions section who carried out the interviews, when, and whether the interviewers were randomly allocated to the interviewees (if not, the possibility of bias should be acknowledged, even if no statistical analysis of results was performed).

3. Authors state that: 

"The quality of the strategies varied in the ability to minimise risk of bias being introduced and highlight that blinding is not simply a 'yes, it is blind' or 'no, it is not.' The strategies exist on a spectrum of effectiveness. Ideally blinding needs to not only conceal the intervention information (e.g. drug X) but also the experimental group membership (i.e. which animals are grouped together in group 1, group 2 etc) during allocation, conduct and outcome assessment"

While it is hard to disagree with this statement, it does however raise an issue in regard to the quality of information provided in systematic reviews trying to assess quality of research, in which blinding is often neither categorized into the categories described by the authors, nor described in a nuanced way, and rather reported as a "yes/no" dichotomy. I would indeed appreciate seeing this issue discussed, particularly considering the connection of the authors to the practice of carrying out systematic reviews. 

4. I am not convinced by the authors' statement that "[lack of blinding] can then lead to animals being handled or data collected more consistently within each experimental group which will reduce the variability seen and could potentially increase the statistical significance of a differences between experimental groups. The argument is presented in a quite diffuse and convoluted manner, and does not provide any references to back it up. It also, in my view, misconstrues what "statistical significance" means. There is moreover a typo ("of a differences"). I agree that the biases resulting from lack of allocation concealment and blinded outcome assessment can lead to an overestimation of effects, particularly because there is evidence in this regard, yet the argument that it reduces variability in such a way that analysis of results will yield smaller p-values, aside giving p-values more importance than they deserve, has not, to my knowledge, been established in the literature. 

5. The authors refer profusely to the ARRIVE guidelines on the matter of blinding of experiments, in a manuscript where the focus is how researchers plan and carry out their experiments, rather than their reporting. Either the authors explicitly make the case that reporting guidelines (theirs in particular or reporting guidelines in general) can be reversed-engineered to plan an experiment, or should be more cautious with such claims. No mention of guidelines on planning of experiments, such as the PREPARE, are made. The latter is particularly unfortunate, since the communication and planning warranted to orchestrate the combined efforts necessary to make multiple persons carry out different, blinded, tasks, is better addressed by the PREPARE. The authors should have realised this upon stating that "The recommended approach of a common welfare strategy is a significant shift and will require exploration of the topic with all the stakeholders to agree the welfare assessment protocol going forward".

6. The ethical issues arising from non-disclosure of treatments (in the broadest sense of the term "treatment", which could well mean "tumour implantation") to animal care staff is underexplored. Knowing the interventions informs animal care and clinical decisions, and the example given by authors of a rat showing piloerection and hunched posture being treated differently according to the expected outcome is a good example. It makes a world of difference in preventing unnecessary suffering that caretakers and vets know whether a clinical sign is a transient state or the first step towards irreversible decline in health and welfare, which may depend on the assigned treatment (or lack thereof). In the context of understaffed animal facilities, allocating manpower for more frequent monitoring of the animals that are more likely to quickly progress to severe stages indeed makes sense and allows for earlier humane endpoints, and thus for more humane science. The impact of different professional objectives (carry out research vs. caring for animals, even between scientific vs. deontological duties) may lead to conflicts between scientists and animal care staff, which must be acknowledged, discussed, and sorted beforehand. The ARRIVE (and, to be fair, almost any other set of reporting guidelines) provide little to no guidance on how to accomplish this. 

7. Another underexplored issue is how abiding to the standards proposed in this manuscript is dependent on the available resources. I know cases where the planner of experiments, executer of said experiments, caretaker (including being responsible for cage change), data manager, and analyst are one and the same person. This might be an extreme scenario, but not that uncommon, even in Europe. This economical bias is even harder to tackle than the biases the authors aim to address, and it may exclude science - and scientists - from lower-income countries from publishing in journals with higher visibility, or at all. This is not a criticism of the authors' call for higher standards, but rather a statement of fact, which I suggest the authors reflect upon in their discussion.

8. The authors are right in acknowledging that fear of errors from blinding is not entirely ungrounded. It could be added that the chances of errors increase with the number of people involved and the number of blinding (and un-blinding) steps necessary.

9. I disagree with the authors' claim that support staff is not "emotionally invested" in the research outcome. Indeed, when research support staff are acknowledged to be an important integral part of the broader institutional research team and hence of the research success (and they should be), within the establishments where they see themselves as valuable and valued, they are indeed invested that experiments yield "positive/good/meaningful" results. To disregard them as either not caring or not understanding the research objectives is quite misguided and tone-deaf, and indeed incompatible with a culture of care. The authors are hence invited to reflect upon whether it is not more effective to refer to blinding of the support staff as a process where the latter are involved and the reasons for why they are blinded to treatment allocation and other features are openly discussed. If not for any other reason, because all humans are hardwired to find patterns and connections (even non-existing ones) anyway, regardless of being "invested" or otherwise, which could lead to performance bias and skewed results.

10. The authors refer to the practice of allocation concealment as "blinding during the allocation and intervention". It seems as it is to follow the description used in the ARRIVE, where different measures carried out for different reasons are neatly gathered under the same category of "blinding", perhaps to fit into the Essential 10 (who does not love a round number?). I am aware that the authors acknowledge the term "allocation concealment" as synonymous to their wordy terminology, but using the same term "blinding" for several contexts may confuse some readers, especially those more familiarized with e.g. the Cochrane terminology. A very brief discussion (i.e. one sentence, perhaps in the abstract/introduction)) on how the concept of "blinding" presented in this manuscript is broader and includes a number of steps that may be referred to by other terms in other references may be useful. 

11. The re-analysis of an un-blinded set of data upon a first round of "unsuccessful" tests (i.e., in the view of many, when p>0.05) is malpractice and should be called for what it is: p-hacking, accompanied by the necessary references. 

12. I understand the need to refer to the ARRIVE at every opportunity (and the conflict of interests is duly acknowledged in the respective section), but this was submitted as a tentative academic paper and not a NC3Rs pamphlet. The constant emphasis given to reporting guidelines on what should be a matter of planning and execution to prevent unwanted errors and biases - for which there are much better guidelines and references in the literature available, most of which are not even mentioned - should make the authors pause and reflect on whether they are not tooting their own horn just a tad too much. Examples include:

- "The inclusion of blinding within the essential set of the ARRIVE guidelines (the Essential 10) [12], and the request to report on all four stages of an experiment is part of the journey to normalise this expectation."

- "The engagement of journals or professional bodies with these quality markers [the ARRIVE] will assist in the culture shift that blinding should be a standard practice."

13. I do not have access to the raw data, but I expect it to be available (duly anonymized). 

14. Minor issues:

* Change "For most the studies" to "For most of the studies"

* Change "consider oncology studies" to "considering oncology studies"

* The authors refer to a "Culture of Care", yet there is no reference directing readers to understand what this is about.

* Where the authors refer to the Irwin behavioural screening assay, a reference on this assay should be added.

* I would suggest in table 4 to divide the third column (Strengths and weaknesses) into two columns, or each cell divided longitudinally into "strengths"/"weaknesses")

---

## [Decision Letter · Decision Letter 2]

23 Sep 2022

Dear Natasha,

Thank you for your patience while we considered your revised manuscript "The reality of masking (blinding) during in vivo research: a qualitative study exploring engagement, barriers, and strategies to move forward" for publication as a Meta-Research Article at PLOS Biology. This revised version of your manuscript has been evaluated by the PLOS Biology editors, the Academic Editor, and two of the original reviewers.

Based on the reviews, we are likely to accept this manuscript for publication, provided you satisfactorily address the remaining points raised by the reviewers. Please also make sure to address the following data and other policy-related requests.

IMPORTANT. Please attend to the following:

a) Please change the Title to something like "A qualitative study of barriers and solutions to blinding researchers to experimental conditions during in vivo research"

b) Please address the remaining points from the reviewers. Regarding the two somewhat editorial points raised by reviewer #3, we think that in both cases you need to strike a pragmatic balance between correctness and comprehensibility, and the previous round may have resulted in an overshoot. In particular, we think you need to adjust the balance on the blinding/masking issue. "Blinding" is clearly the most widely understood term for this procedure, and you should use this alone in the Title (see point "a"). At the beginning of the Abstract, you should then reverse the terms, thus: "In animal experiments, blinding (also known as masking)...," and then largely use "blinding" in the Abstract. You then correctly explain in the Intro why we should be seeking to move away from the use of the ableist term "blinding"; from that point onwards, it would be fine to use "masking" throughout, but perhaps with the occasional reminder to the reader that this is what is currently widely known as "blinding." Anyway, we leave most of this to your discretion, but ask that "blinding" be the sole term used in the title.

c) We note that in the funding declaration you say “The authors received no specific funding for this work.” Please can you confirm that this is indeed the case?

We expect to receive your revised manuscript within two weeks. 

*Published Peer Review History*

*Press*

Sincerely,

Roli

Roland Roberts, PhD

Senior Editor,

rroberts@plos.org,

PLOS Biology

REVIEWERS' COMMENTS:

Reviewer #1:

[identifies herself as Nicole C. Nelson]

The revised version of the manuscript has a much more complete description of the study's methodology. There are a few points where I think further clarification would be helpful. 

1. Even though the study was not designed to unpick whether career stage/status or other demographic variables impacts attitudes towards blinding, it would still be helpful to both report these and to have at least some commentary on which variables seemed to track with differences in the barriers identified - e.g., did industry researchers/senior researchers/staff members seem to be more attuned to some kinds of constraints than others? Likewise, it is still useful to report on which themes came up a lot and which came up rarely even if prevalence can't be estimated with more precision. One of the ways that an exploratory study like this one can add value is by helping future researchers get a sense of what the relevant sources of variation are in this area so that they can make a more educated guess about what factors they should control for in their study design, and as written at present the manuscript does not give a lot of information of that kind.

2. The claim in the limitations section that the sample size should be sufficient to identify the majority of the barriers researchers might encounter doesn't seem justified to me. In the study cited in support of this claim (Faulkner et al 2003), the original authors are looking at how many people are needed to find problems in a single application (an employee timesheet), so all of the variation is in the positionality of the users. In the present study, differently positioned users are talking about different experiments, so there's variation on both ends. 

3. Reporting on data saturation could help with the problem identified above in (2): since the development of the coding scheme was iterative, the authors could report on how many interviews were needed to develop a reasonably stable coding scheme (that is, to reach saturation). If the authors reached saturation using only a subset of the articles, then I'd have more confidence in the claim that the majority of possible themes have been identified. For an example of this methodology, see Guest, Bunce, and Johnson (https://doi.org/10.1177%2F1525822X05279903). This article has also been very widely cited as evidence that ~12 interviews per group is generally enough to reach saturation, although what counts as a group (as discussed in 1 above) is a key question to be worked out!

4. One of the other reviewers notes in their original report that it's ironic that a study on masking such as this one cannot itself be masked, but masking during data analysis is certainly possible. I'm not sure it would make a huge difference for this particular study (or how possible it would be given the pre-existing relationships between interviewers and interviewees), but given that this is a manuscript on masking it would probably be good to be precise about this in the methods section. Right now the methods section states that the data was "stripped of any identifying information," and it's not clear if this was simply de-identification of the type that the IRB would require to conceal participants' identities from members outside the study team, or if the authors were attempting to mask themselves so that they would be unaware which interview summary they were dealing with when developing and applying the coding scheme. 

In terms of the analysis, I have a few points.

1. First, one minor point: The sentence introducing the paragraph discussing the case of transient piloerection is unclear, in that I wasn't sure what review the authors were referring to until I went back and read the other reviewers' comments in more detail. It's a very useful example to include, but it should be clear that it's an example coming from a reviewer and not from a review of the interview data.

2. And now, the major point: Throughout the results section, it is difficult to tell whether the authors are reporting on what interviewees told them or making recommendations about what best practice is. For example, in the very first paragraph of the results section, the manuscript states that masking was not relevant for a number of the experiment types included—is this the authors' determination, or the interviewees' assessment? Likewise, did participants agree that in 28/30 cases that masking could be implemented in some stages to reduce bias? Did participants rank different masking strategies as low/moderate/high quality? Did participants express concern that power dynamics or a want to please might decrease the effectiveness of bias reduction strategies? Some rewriting is needed here so that readers can easily identify which elements are opinions held by the participants and which ones are judgements/recommendations from the authors.

3. If it is the case that many of these assessments about the quality of various strategies etc are actually the authors' judgements and not reports of participants' assessments, then I think the authors should be much more upfront about the fact that they are offering their opinions on what counts as a high/low quality strategy, rather than offering guidelines of the type that would be more typically developed through a broad consultative process. This seems especially important given that many of the authors have been involved in the production of the ARRIVE guidelines and so readers might be more inclined to read what's written here as a guideline rather than as an exploratory study that provides a starting point for future discussion (which I think is what the authors intend). 

Reviewer #3:

The authors have made a considerable effort to reply to the reviewers' comments, and were able to address most of them in a meaningful way. I have divergences of opinion with them, but this should not disqualify an academic paper from being published, in my view.

I have two final comments, that I will leave to the authors' - or editor's - judgement whether to consider or not.

1. The authors decided to accommodate the other reviewer's suggestion to change "blinding" to "masking", even though it is not standard terminology (e.g. one does not usually refer to "double masking"). I'm not surprised to see this sort of request, in this day and age, and though certainaly well-intentioned, I find it utter nonsense. Perhaps this is because English is not my mother tongue. So I do find it odd that representatives of an organisation that does not use this terminology (vide the ARRIVE guidelines themselves) would be so quick to change it upon request, despite adding to the already confusing semantic chaos, not being the term used in the interviews, or registered in the notes taken, or originally used in the manuscript (as the wording now used in the sentence justifying this choice seems to suggest, which I couldn't help connect with the practice of HARKing). I find it nonsensical but if the authors can live with that decision, who am I to disagree? 

2. Changing p-value to "statistical significance" does little to make sense of this claim: "This can then lead to animals being handled or data collected more consistently within each experimental group which may introduce bias and could potentially increase the statistical significance of a difference between experimental groups." It is not the "statistical significance" we are affecting, but rather the potential direction or magnitude of differences in the data collected by changing how animals "perform" (hence 'performance bias'). A p-value is a conditional probability of the observed differences in the data being as extreme or more if there were no effect, and it is affected not only by differences between groups, but also the variability in the data, and the statistical power. I understand what the authors are trying to say, but referring to 'statistical significance', here, is imprecise. However, wrong as it may be, it is not a hill I wish to die on, as I would not want to revise this manuscript yet again, so I will leave it to the editor's judgment.

---

## [Editor Report · Decision Letter 3]

7 Oct 2022

Dear Natasha,

Thank you for the submission of your revised Meta-Research Article "A qualitative study of the barriers to using blinding in in vivo experiments and suggestions for improvement." for publication in PLOS Biology. On behalf of my colleagues and the Academic Editor, Cilene Lino de Oliveira, I'm pleased to say that we can in principle accept your manuscript for publication, provided you address any remaining formatting and reporting issues. These will be detailed in an email you should receive within 2-3 business days from our colleagues in the journal operations team; no action is required from you until then. Please note that we will not be able to formally accept your manuscript and schedule it for publication until you have completed any requested changes.

Sincerely,

Roli

Senior Editor

PLOS Biology

rroberts@plos.org